# Ethyl Formate Fumigation against Pineapple Mealybug, *Dysmicoccus brevipes*, a Quarantine Insect Pest of Pineapples

**DOI:** 10.3390/insects15010025

**Published:** 2024-01-02

**Authors:** Tae Hyung Kwon, Dong-Bin Kim, Bongsu Kim, Joanna Bloese, Byung-Ho Lee, Dong H. Cha

**Affiliations:** 1United States Department of Agriculture-Agricultural Research Service, Pacific Basin Agricultural Research Center, Hilo, HI 96720, USA; 2Oak Ridge Institute for Science and Education, Oak Ridge, TN 37831, USA; 3College of Tropical Agriculture and Human Resources, University of Hawaii at Manoa, Hilo, HI 96720, USA; 4Institute of Quality & Safety Evaluation of Agricultural Products, Kyungpook National University, Daegu 41566, Republic of Koreabyungholee@hotmail.com (B.-H.L.); 5Animal and Plant Quarantine Agency, Gimcheon 39660, Republic of Korea

**Keywords:** pineapple mealybug, *Dysmicoccus brevipes*, ethyl formate, methyl bromide alternative, phytosanitation

## Abstract

**Simple Summary:**

Pineapple mealybug, *Dysmicoccus brevipes*, is a major pest of pineapple production and trade barrier. Methyl bromide fumigation (MB) has generally been used to disinfest imported pineapples. However, its use has been phased out due to its impact on the ozone layer and human health. As a first step to developing MB alternative treatment for imported pineapples, we evaluated whether ethyl formate (EF) fumigation could be an effective disinfestation treatment for pineapple mealybug. In a scaled up trial, EF fumigation with 70 g/m^3^ EF for 4 h at 8 °C with 20% pineapple loading ratio (*w*/*v*) resulted in complete control of pineapple mealybugs treated with no apparent negative impact on pineapple quality. Our results suggest that EF fumigation is a potential disinfestation treatment for pineapple mealybug in imported pineapples.

**Abstract:**

Pineapple mealybug, *Dysmicoccus brevipes* (Hemiptera: Pseudococcidae), is a significant pest in pineapple production and a key trade barrier. We explored the potential use of ethyl formate (EF) as a methyl bromide alternative for the postharvest fumigation of *D. brevipes* in imported pineapples. When treated at 8 °C for 4 h, EF fumigation was effective against *D. brevipes* with LCt_99,_ the lethal concentration × time product of EF necessary to achieve 99% mortality of *D. brevipes* nymphs and adults at 64.2 and 134.8 g h/m^3^, respectively. Sorption trials conducted with 70 g/m^3^ EF for 4 h at 8 °C using 7.5, 15 and 30% pineapple loading ratios (*w*/*v*) indicated that loading ratio lower than 30% is necessary to achieve the LCt_99_ values required to control *D. brevipes*. In a scaled up trial using 1 m^3^ chamber, EF fumigation with 70 g/m^3^ for 4 h at 8 °C with 20% pineapple loading ratio (*w*/*v*) resulted in a complete control of *D. brevipes* treated. There were no significant differences in hue values, sugar contents, firmness, and weight loss between EF-treated and untreated pineapples. Our results suggest that EF is a promising alternative to methyl bromide fumigation for the postharvest phytosanitary disinfection of *D. brevipes* in pineapples.

## 1. Introduction

Pineapple (*Ananas comosus*) is a highly traded fruit with significant economic value, serving as a popular fresh commodity in many countries. Its global market value has been estimated to surpass $2.5 billion [1]. Pineapple is cultivated in many countries, with major producers including Brazil, Costa Rica, Hawaii, Indonesia, Philippines, and Thailand [2]. In terms of imports, the major markets for fresh pineapples include the United States, the European Union, China, Canada, Japan, and the Republic of Korea [3,4]. 

One of the major pests of pineapples is the pink pineapple mealybug, *Dysmicoccus brevipes* (Cockerell, 1893) (Hemiptera: Pseudococcidae) [5]. It is a serious threat to pineapple production in its native ranges of Central and South America [6] and in invaded regions, such as Hawaii [7], Sri Lanka [8], and China [9]. *Dysmicoccus brevipes* go through three nymphal stages before becoming adults and do not lay eggs. Instead, the adult female parthenogenetically produces eggs that hatch inside her body and gives birth to fully formed live nymphs (i.e., ovoviviparous) [10,11]. Feeding on young growth and roots, *D. brevipes* can cause considerable damage that leads to reduced yields and significant economic losses [12]. Moreover, *D. brevipes* attacks a wide range of agricultural, horticultural and forest species [13] and vectors the pineapple mealybug wilt-associated virus that can have a devastating impact on pineapple crops [14] and >30 potential horticultural and ornamental hosts with economic importance [15,16]. This makes *D. brevipes* a serious trade barrier for pineapples [17]. Thus, some of the major pineapple-importing countries, such as Korea and Japan, have implemented strict quarantine regulations mandating disinfestation treatment of *D. brevipes* upon interception from the imported pineapples [18]. 

Methyl bromide (MB) has traditionally been used for the phytosanitary treatment of imported pineapples [19]. For example, in Korea, MB treatment at 64, 40 and 24 g/m^3^ at 4.5–10, 15–21 and >26 °C, respectively, for a duration of 2 h has been approved to disinfest imported pineapples based on MB efficacy against *Planococcus citri* [20]. However, due to its negative impact on ozone depletion and human health risks, most MB uses have been phased out since 2005 except for quarantine and preshipment purposes under a critical use exemption [21,22,23] and, in particular, MB use for tropical fruit quarantine treatment has been banned in Korea since 2021 [24]. Although a combination treatment of ethyl formate (EF) and phosphine (PH_3_) of 1.0 g/m^3^ PH_3_ and 25.1 g/m^3^ EF for 4 h at 8 °C has replaced the MB treatment and is currently being used to disinfest imported pineapples in Korea, the combination treatment is more costly and logistically challenging than a stand-alone treatment of EF or PH_3_ [25]. In addition, the MB and EF + PH_3_ treatments are established based on their efficacy on *P. citri*, due to a lack of access to *D. brevipes* for treatment development [26]. Thus, there is a need for a treatment specifically targeting *D. brevipes*. 

In this study, we evaluate the potential of EF fumigation as an MB alternative stand-alone treatment for *D. brevipes*. EF is a naturally occurring chemical that can be found in oranges, cheese, and stored grain [27]. Its effectiveness as a fumigant has been demonstrated against a wide variety of surface pests, such as mealybugs and scale insects, that infest various fruit crops and nursery plants [28,29,30,31]. EF is generally recognized as safe (GRAS) [32,33] and readily breaks down into ethanol and formic acid with minimal residue [34]. However, EF has relatively high sorption to fresh commodities [25,35,36] and can cause phytotoxicity depending on the concentration of EF [37] and species/cultivars of fresh commodities treated [28,30]. Here, we report the efficacy of EF as a disinfestation treatment against *D. brevipes*. Specifically, we (1) determined EF efficacy on nymphal and adult life stages of *D. brevipes*, (2) evaluated EF sorption in pineapples, (3) conducted a large-scale trial using a 1 m^3^ chamber, and (4) evaluated the effect of EF fumigation on the quality of pineapples. 

## 2. Materials and Methods

### 2.1. Insects and Pineapples

*Dysmicoccus brevipes* were collected from infested pineapple roots at the Dole Plantation in Wahiawa, Hawaii. To establish the colony, *D. brevipes* nymphs hatched from the field-collected adults were transferred to fresh organic Kabocha squash [38] and kept at the U.S. Pacific Basin Agricultural Research Center in Hilo, Hawaii under 25 ± 1 °C, 80 ± 10% relative humidity, and a 16:8 L:D cycle. EF efficacy trials (see below) were conducted using *D. brevipes* nymphs and adults from the colony. Scaled up trials (see below) were carried out using both colony *D. brevipes* and field-collected *D. brevipes*. Pineapples were purchased from local groceries and stored at 8 °C until they were used for EF sorption, large-scale fumigation, and quality evaluation tests.

### 2.2. Ethyl Formate 

Liquid ethyl formate (EF, purity: 97%) was purchased from Sigma Aldrich Co. (St. Louis, MO, USA). For efficacy trials, liquid EF was applied on a filter paper (90 mm dia., Whatman, Inc., Buckinghamshire, UK) for vaporization. For the scaled up (1.0 m^3^) trials, the liquid EF was vaporized using a prototype EF vaporizer (Safefume Inc., Daegu, Republic of Korea) and EF gas was propelled into the fumigation chamber using nitrogen gas from a cylinder [39]. 

### 2.3. Determination of Ethyl Formate Concentration and Ct (Concentration × Time) Product 

Ct products of EF in fumigation chambers were calculated by determining the concentrations of EF at 0.2, 2 and 4 h after exposure to EF. A portable Agilent 990 Micro gas chromatograph (GC) equipped with a thermal conductivity detector (TCD) after separation on a PoraPLOT Q Column (10 m × 0.25 mm i.d., 8 μm film thickness; Agilent technology, Santa Clara, CA, USA) was used for the determination. GC oven temperature was continuous at 80 °C. The temperatures of the injector and detector were 100 and 180 °C, respectively. Helium was used as a carrier gas at the flow rate of 1.5 mL/min. The concentration of EF was determined using standard curves generated using external EF standards prepared by spiking a known volume of liquid EF into a 1 L Tedlar^®^ gas sampling bag (SKC Inc., Eighty-Four, PA, USA). The Ct products were calculated as follows: $Ct=\sum \frac{\left(C_{I}+C_{i+1}\right)(t_{i+1}+t_{i})}{2}$, where C = concentration of fumigant (g/m^3^), t = time of exposure (h), i = order of measurement, and Ct = concentration × time product (g/h m^3^) [40].

### 2.4. Efficacy of Ethyl Formate Fumigation against Dysmicoccus brevipes Nymphs and Adults 

EF efficacy trials were conducted using 2.4 L desiccators (Corning Inc., Corning, NY, USA) [41]. Briefly, liquid EF was injected into the desiccator through a rubber septum installed on a glass stopper, using a 500 μL gas-tight syringe (Hamilton Inc., Reno, NV, USA), following a scheduled dose calculated to target the tested dose range between 5.0 and 70.0 g/m^3^ (e.g., 5, 10, 20, 30, 40, and 50 g/m^3^ for nymphs; 20, 30, 40, 50, 60, and 70 g/m^3^ for adults). A 90 mm diameter filter paper (Whatman, Inc., Buckinghamshire, UK) was inserted into the glass stopper for the evaporation of injected EF. The precise volume of each desiccator was measured as the weight of the water required to fill the desiccator at 21 °C. Each desiccator had a magnetic stirrer at the bottom to facilitate circulation of the fumigant. The nymphs and adults for efficacy trials were prepared by placing a piece of the squash infested with three adult *D. brevipes* on a new organic Kabocha squash and allowing the adult *D. brevipes* to give birth to live nymphs, which were reared to second and third instar nymphs or adults. For each trial, a screened Petri dish (120 × 80 mm) containing a slice of infested squash with approximately 30 nymphs (mostly third instar) or 20 adults was placed in a fumigation desiccator. A total of 630 nymphs and 720 adults were tested for efficacy trials. Fumigation trials were conducted at 8.0 ± 0.5 °C for 4 h in low-temperature incubators (B.O.D. incubator, VWR International, West Chester, PA, USA). 

Following the completion of fumigation, the desiccators were subjected to 1 h aeration process in a fume hood. Subsequently, the treated nymphs and adults were removed from the desiccators and transferred to the rearing room, maintaining conditions at 25 ± 1 °C and 80 ± 10% relative humidity, with a 16:8 L:D cycle. Nymph and adult mortalities were assessed 3 and 7 days postfumigation, respectively, through careful probing and visual examination of leg or appendage movement upon brush contact. To account for mortalities unrelated to the fumigation, untreated control nymphs and adults were included in each trial. Corrected mortality was computed using the Abbot formula [42]: corrected mortality = [(% treatment mortality − % control mortality)/(100 − % control mortality)] × 100, with control mortalities for nymphs and adults at 5.6% and 2.2%, respectively. Each treatment and control were replicated five times.

### 2.5. Assessment of Ethyl Formate Sorption in Pineapples 

The sorption of EF in pineapple was evaluated at Kyungpook National University in Daegu, Korea, using a 0.275 m^3^ fumigation chamber. Three different loading ratios, 7.5, 15.0 and 30.0% (*w*/*v*), were tested. EF was applied at 70 g/m^3^ for 4 h at 8.0 ± 0.5 °C using a prototype EF vaporizer (Safefume Inc., Daegu, Republic of Korea) with nitrogen gas as a propellent. To enhance EF gas circulation within the chamber, a fan was positioned at the bottom. Post EF treatment, concentrations of EF inside the fumigation chamber were determined at 0.2, 2, and 4 h by extracting EF gas from both inside and outside pineapple boxes, using gas sampling bags (SKC Inc., Eighty-Four, PA, USA) and a gas sampling pump. The sorption rate was determined based on the reduction in concentration over time using the equation (C_n_/C_0_), where C_n_ = concentration measured at a given hour, C_0_ = initial concentration. Additionally, Ct products of EF were calculated to estimate the loading ratio required to achieve the Ct product associated with LCt_99_.

### 2.6. Scaled Up Ethyl Formate Fumigation for Dysmicoccus brevipes Control 

Scaled up fumigation trial was conducted using a 1.0 m^3^ stainless steel fumigation chamber with a clear acrylic door (0.91 × 0.91 × 1.20 m) situated in an outdoor walk-in cooler (Polar King International Inc., Fort Wayne, Indiana) at Pacific Basin Agricultural Research Center in Hilo, Hawaii. EF was treated at 70 g/m^3^ for 4 h at 8 °C with a 20% loading ratio (*w*/*v*) of pineapples (9 box, 6 pineapples/box) procured from a local retailer. In this scaled up trial, 1836 *D. brevipes* (1043 nymphs and 793 adults) from the laboratory colony and 278 wild *D. brevipes* (170 nymphs and 108 adults) were subjected to treatment. Colony *D. brevipes* were obtained by excising a section of Kabocha squash infested with *D. brevipes* nymphs and adults, then gently brushing off first and second instar nymphs using a paintbrush, so mostly adult *D. brevipes* and third instar nymphs remained on the squash piece. Around one hundred colony *D. brevipes* adults and 150 nymphs were placed in each of three separate insect breeding dishes (947 mL deli cup with screened lid). These dishes were distributed inside pineapple boxes in the bottom, middle, and top sections of the fumigation chamber. Wild *D. brevipes* adults and nymphs were sourced from infested pineapple roots collected at the Dole plantation. Three insect breeding containers (9.4 L) filled with pineapple roots infested with wild *D. brevipes* nymphs and adults were positioned at the bottom, middle and top sections of the fumigation chamber. A similar number of untreated colony and wild *D. brevipes* were also tested in an untreated control scaled up trial to account for mortality not directly linked to the fumigation treatment. Nymph and adult mortalities were assessed at 3 and 7 days postfumigation, respectively, as detailed earlier. Due to the ovoviviparous nature of *D. brevipes*, egg mortality was indirectly inferred from the emergence of first instar nymphs during the 15 d period following the fumigation treatment [43]. All of the fumigated adults (793 colony adults and 108 wild adults) and a small number of randomly sampled untreated control adults (20 colony adults and 6 wild adults found on one of the Kabocha squash pieces and one of the pineapple roots used in untreated control trial) were evaluated. The EF concentration was measured at 0.2, 2.0, and 4.0 h post EF treatment from the inside of pineapple boxes at the top, middle and bottom parts of the fumigation chamber, as well as from outside pineapple boxes at the middle part of the fumigation chamber. EF gas was drawn into 1 L gas sampling bags using an air pump through gas sampling silicon tubes. The concentration of EF and Ct products was determined as described earlier. Corrected mortality was calculated as previously described, with control mortalities of colony nymph, wild nymph, colony adult, and wild adult at 18.7%, 15.3%, 16.5% and 16.4%, respectively.

### 2.7. Effect of Ethyl Formate Fumigation on Pineapple Quality

The assessment of EF treatment effects on pineapple quality was carried out at Kyungpook National University in Daegu, Korea, using imported pineapples purchased from a local grocery. EF fumigation treatment was conducted in three 0.275 m^3^ fumigation chambers, following the previously described methods for the scaled up trial. Pineapples were stored at 8 °C for 3 d before the trial and five pineapples were treated per each chamber. Subsequent to the 4 h EF-treated and untreated control trials’ completion (n = 3), the fumigation chambers were vented for 1 h at 8 °C. Following ventilation, one pineapple was randomly selected from each chamber (e.g., 3 EF treated and 3 untreated pineapples) and kept at 8 °C until the quality evaluation. After a 7 d storage period, the effect of EF treatment on pineapple crown color, fruit soluble sugar content, fruit firmness, and weight loss was evaluated. Pineapple crown color was evaluated by determining hue values using a colorimeter (TES 135A, Electrical & Electronic Corp., Taipei, Taiwan). For measuring sugar content (% Bx), pineapple juice was extracted from 100 g pineapple pulp and analyzed with a portable refractometer (Hand refractometer ATC-1E, Atago Co., Ltd., Tokyo, Japan). Fruit firmness was assessed by pushing an 8 mm steel plunger to the center of the pineapple fruit using a firmness tester (53205 Digital fruit firmness tester, TR Turoni, Foril, Italy). Weight loss was calculated by comparing the weights of pineapples before and 3 days after fumigation for both EF-treated and untreated groups.

### 2.8. Statistical Analyses 

Probit analysis [44] was employed to assess the efficacy of EF on the mortality of *D. brevipes*. This analysis involved determining the slopes of the probit transformations and conducting chi-square tests to assess data homogeneity across various treatments. Differences in pineapple crown color, soluble sugar content, firmness, and weight loss between EF-treated and untreated pineapples were examined using an independent *t*-test (SAS ver. 9.4). 

## 3. Results

### 3.1. Ethyl Formate Efficacy against Dysmicoccus brevipes 

Adult *D. brevipes* appeared more tolerant to EF treatments than nymphs. Numerically, the LCt_99_ value for adults was more than two times greater than the LCt_99_ value of nymphs with the LCt_99_ of adults and nymphs at 134.8 and 64.2 g h/m^3^, respectively (Table 1). A similar trend was also observed for LCt_50_ values with the LCt_50_ of adults and nymphs at 13.5 and 8.3 g h/m^3^, respectively. 

### 3.2. Assessment of Ethyl Formate Sorption under Different Pineapple Loading Ratio 

After EF is injected into the fumigation chamber, the concentrations of EF decreased over time both inside and outside of pineapple boxes. A greater level of concentration reduction (i.e., sorption) resulted from a greater pineapple loading ratio (outside pineapple box: Y = −168X + 211.1, R^2^ = 0.9819, *p* = 0.0001; inside pineapple box Y = −372.5X + 217.7, R^2^ = 0.9819, *p* = 0.0001, where Y = Ct product of EF, X = loading ratio of pineapples). In addition, EF sorption was greater on the inside than on the outside of the pineapple boxes. For example, 4 h after EF injection, the concentration of EF outside the pineapple box was reduced by 48, 59 and 62% at 7.5, 15.0 and 30.0% loading ratios, respectively, while the concentration of EF inside the pineapple box was reduced by 58, 65 and 80%, respectively (Figure 1A). As the Ct product inside the pineapple box at a 30% loading ratio (103.0 g h/m^3^) was lower than the LCt_99_ value required for adult *D. brevipes* (134.8 g h/m^3^; Figure 1B), we used a 20% loading ratio in subsequent scaled up trials. 

### 3.3. Scaled Up Ethyl Formate Fumigation for Dysmicoccus brevipes Control 

EF treatment for 4 h at 8 °C resulted in complete control of 901 adults (793 colony and 108 wild) and 1213 nymphs (1043 colony and 170 wild). The EF Ct values measured from inside and outside pineapple boxes in different locations of the fumigation chamber were all greater than the LCt_99_ value (134.8 g h/m^3^) for *D. brevipes* adults (Table 2). When EF-treated or untreated control adults were monitored over 15 days, no nymphs emerged from all of the EF-treated adults (793 colony and 108 wild adults), while 66 and 23 first instar nymphs emerged from randomly sampled 20 colony and 6 wild *D. brevipes* adults not treated with EF (Table 2).

### 3.4. Effect of Ethyl Formate Fumigation on Pineapple Quality

EF fumigation did not affect pineapple quality at the EF level used in the scaled up trial. When pineapples were stored at 8 °C over 7 d after the trials, there were no significant differences in hue values of pineapple crown, fruit firmness, sugar content, and weight loss between EF-treated and untreated pineapples (Table 3). 

## 4. Discussion

Our results demonstrated that EF fumigation is an effective stand-alone treatment against *D. brevipes* infesting pineapples. In a scaled up trial, EF treatment at 70 g/m^3^ for 4 h at 8 °C with a 20% pineapple loading ratio (*w*/*v*) resulted in complete control of 901 *D. brevipes* adults treated, which were two times more EF tolerant than nymphs. The complete control of *D. brevipes* was achieved without noticeable changes in pineapple quality, with no significant differences in sugar content, hue value, firmness, and weight loss between the untreated control vs. EF pineapples. Together, these results support EF as a potential MB alternative phytosanitary treatment of *D. brevipes* in pineapples. 

It is well known that hydrolysis, condensation, and/or sorption of EF result in a decreased concentration of EF inside a fumigation chamber once EF is injected [45,46,47]. This pattern was consistent in our study, with EF concentration decreasing over time in the headspace of the fumigation chamber. The reduction was stronger with a 30% pineapple loading ratio (*w*/*v*) than lower levels of loading ratio (15% and 7.5%). In addition, within the same loading ratio, the concentration of EF inside the pineapple box was significantly lower than the concentration outside the box, which was consistent with EF treatments for other commodities, such as mushrooms and sweet persimmons [48,49]. Importantly, the concentration × time product of EF determined inside pineapple boxes with a 30% loading ratio was lower than the 134.8 g h/m^3^ target necessary for the LCt_99_ level control of *D. brevipes* adults. Thus, to ensure that the concentration of EF was maintained at the target level inside the pineapple box, we adjusted the pineapple loading ratio to 20% (*w*/*v*) for the scaled up trial, which resulted in 178.8 g h/m^3^ EF inside the box and complete control of treated *D. brevipes* adults and nymphs. This indicates that careful consideration of logistical and operational constraints is necessary to ensure meeting the designed target treatment level in commercial practices [50]. 

Previous studies have shown the effectiveness of EF fumigation against a wide range of insect pests, and this study adds one more example of EF fumigation for surface pests [51]. In addition to its effectiveness on many target pests, one advantage of EF fumigation is the lack of harmful residues. EF breaks down readily to ethanol and formic acids [52,53], is a flavoring agent, and is generally regarded as safe [35]. One of the limitations of EF is that it can negatively affect the quality of fresh commodities, including discoloration and accelerated decay of fruit [35,51,54]. However, EF phytotoxicity often depends on its concentration and the species and cultivar of treated plants [55,56] and can be minimized by optimizing treatment parameters, such as treatment concentration, time of exposure, and temperature [35,57,58]. Although EF treatment can cause phytotoxicity on pineapples at 100 g/m^3^ EF for 4 h at 8 °C (personal observation, THK), in this study, there was no significant impact on sugar content, crown color, firmness, and weight loss when pineapples were treated with 70 g/m^3^ EF for 4 h at 8 °C. This suggests a need for careful attention to treatment conditions, such as EF dosage, temperature, and duration of treatments. Similar results have been reported for MB fumigation on pineapples. Although no phytotoxicity was observed with 2 h fumigation of 32 g/m^3^ MB at 22.7–34.6 °C or 40 g/m^3^ MB at 20.0–25.0 °C, longer MB treatment durations (4 h and 8 h) based on the same MB dosages resulted in phytotoxicity in pineapples [19].

For *D. brevipes*, determining EF efficacy for the egg stage was not practical as *D. brevipes* mothers birth live nymphs after their eggs hatch inside their body [10,11]. However, in some of the mealybug species that lay eggs (e.g., citrus mealybug), the egg stage has been shown to be the most EF-tolerant [25,28]. Thus, in this study, we monitored all EF-treated and a small portion of untreated adults from the scaled up trials over 2 weeks for the emergence of new nymphs. First instar nymphs emerged only from untreated *D. brevipes* mothers, suggesting EF is an effective treatment to target *D. brevipes* egg stages. However, from this study, it is not clear whether the EF treatment actually killed the eggs or if EF treatment induced adult mortality that somehow affected the birth of live nymphs. 

In conclusion, our study suggests EF as an effective alternative to MB for phytosanitary disinfection against *D. brevipes* in pineapples without a negative impact on pineapple quality. Work will continue to optimize the use of EF for *D. brevipes* control from the perspectives of food security, environmental health, and human health. Specifically, additional studies will be conducted to (1) confirm the effect of EF on *D. brevipes* reproduction using larger samples; (2) conduct commercial scale trials using >30,000 specimens of adults, the most tolerant life stage; and (3) evaluate EF efficacy on *D. brevipes* at different temperature ranges.

## Figures and Tables

**Figure 1 insects-15-00025-f001:**
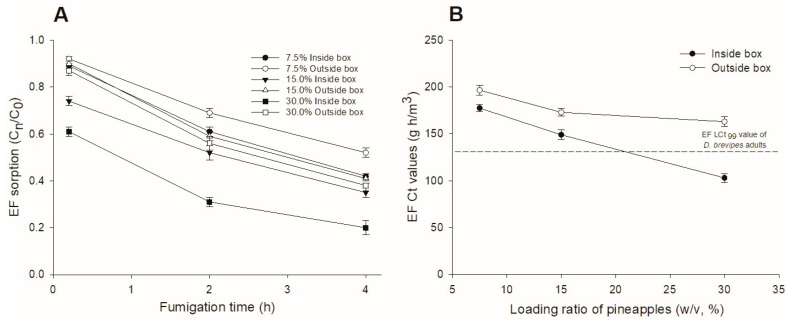
Ethyl formate (EF) (**A**) sorption inside and outside pineapple boxes under different pineapple loading ratios and (**B**) concentration × time products measured inside and outside pineapple boxes under different loading ratios when pineapples were treated with 70 g/m^3^ EF for 4 h at 8 °C with 7.5, 15.0 and 30.0% pineapple loading ratios (*w*/*v*). Dashed line in (**B**) indicates EF concentration × time (Ct) product value necessary for 99% mortality of *Dysmicoccus brevipes* adults (LCt_99_).

**Table 1 insects-15-00025-t001:** Lethal concentration × time (LCt, g h/m^3^) of ethyl formate (EF) for nymph and adult of *Dysmicoccus brevipes* under 4 h EF fumigation at 8 °C. CI: confidence interval, SE: standard error.

Stage	Number Treated	LCt_50_ (95% CI)	LCt_99_ (95% CI)	Slope ± SE	*df*	*χ^2^*
Nymph	630	8.3 (6.7–9.8)	64.2 (46.8–102.0)	2.6 ± 0.3	6	25.2
Adult	720	13.5 (9.7–17.5)	134.8 (84.5–290.0)	2.3 ± 0.3	6	78.2

**Table 2 insects-15-00025-t002:** Ethyl formate concentration × time (Ct) product values inside the pineapple boxes at top, middle, and bottom parts of the fumigation chamber; number of colony and wild *Dysmicoccus brevipes* adults and nymphs tested; mortality (Mean ± SE) of adults and nymphs; and number of first instar nymphs (Mean ± SE) emerged from adults from EF treated (70 g/m^3^ EF for 4 h at 8 °C) and untreated control scaled up trials conducted in 1 m^3^ fumigation chamber with 20% pineapple loading ratio (*w*/*v*). The number of newly emerged first instar nymphs was monitored from all EF-treated adults (793 colony adults and 108 wild adults) and randomly sampled untreated control adults (20 colony adults and 6 wild adults). Corrected mortality of EF-treated *D. brevipes* adults and nymphs was also 100%.

Treatment	Location	Ct Value (g h/m^3^)	Number of Adults		Number of Nymphs		Mortality of Adults (%)		Mortality of Nymphs (%)		Number of Hatched Nymphs	
			Wild	Colony	Wild	Colony	Wild	Colony	Wild	Colony	Wild	Colony
Control	Top	0.0	62	207	102	392	16.4 ± 4.7	16.5 ± 1.2	15.3 ± 1.3	18.7 ± 1.8	23	66
	Middle	0.0	48	270	69	383						
	Bottom	0.0	69	327	92	520						
Treatment	Top	183.0	32	263	60	342	100.0 ± 0.0	100.0 ± 0.0	100.0 ± 0.0	100.0 ± 0.0	0	0
	Middle	176.1	41	289	50	397						
	Bottom	177.4	35	241	60	304						

**Table 3 insects-15-00025-t003:** Effect of ethyl formate (EF) fumigation treatment (70 g/m^3^ for 4 h at 8 °C in 1.0 m^3^ chamber) on pineapple crown hue value, firmness (HB), sugar content (Brix), and weight loss (%) of imported pineapples after 7-day postfumigation period. Different letters on means indicate significant differences between EF-treated and untreated control by *t*-test at *p* < 0.05.

Treatment	Hue Value (Mean ± SE)	Firmness (Mean ± SE)	Sugar Content (Mean ± SE)	Weight Loss (Mean ± SE)
Control	70.8 ± 3.0 ^a^	41.3 ± 0.9 ^a^	14.0 ± 1.2 ^a^	5.1 ± 0.3 ^a^
EF treated	70.2 ± 3.7 ^a^	40.5 ± 1.6 ^a^	14.3 ± 0.7 ^a^	4.5 ± 0.3 ^a^

## Data Availability

The data that support the findings will be made available through a Data Transfer Agreement following an embargo from the date of publication to allow for the commercialization of research findings.

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
