# Peer review of "Ethyl Formate Fumigation against Pineapple Mealybug, Dysmicoccus brevipes, a Quarantine Insect Pest of Pineapples"

_insects, 2024, doi:10.3390/insects15010025_

Round 1
Reviewer 1 Report
Comments and Suggestions for Authors
Dear author, send my suggestions inside of documento, congratulations for their research.

Author Response
Thank you for the comments. We incorporated most of suggestions in the revised manuscript. Also, attached please find the PDF with our point-by-point responses to the comments.

Reviewer 2 Report
Comments and Suggestions for Authors
The present study is about the efficacy of Ethyl formate as a fumigation agent against Pineapple mealybug, Dysmicoccus brevipes, in search of alternatives to methyl bromide fumigation. The study is well conducted, the statistical analysis are correct and the results support the conclusions drawn. Some minor explanations are needed in the material and methods section.
Efficacy of EF fumigation against D. brevipes
How many concentration levels were tested? Which were?
L 144. “…with control mortalities of nymph and adult at 5.6 and 2.2%, respectively “ What are the percentages? The same for L 191
It is not clear how were displaced the colonies into the pineapple boxes. Were the Kabocha squach pieces placed into the boxes? According to these: “Colony adult D. brevipes were obtained by cutting out a piece of Kabocha squash infested with D. brevipes nymphs and adults and by gently brushing off nymphs using a paint-168 brush.”… insects were placed without kabocha squach, but in the following sentence “…(20 colony adults and 6 wild adults found on one of the Kabocha squash piece and one of the pine-apple roots used in untreated control trial)” it seems that kabocha squash was inside the boxes of control treatment. Please, clarifiy.
Author Response
Thank you for the comments and suggestions. Point-by-point responses attached.

Reviewer 3 Report
Comments and Suggestions for Authors
The manuscript describes the use of Ethyl formate as fumigant against the the pineapple mealybug Dysmicoccus brevipes. The achievements can be important for phytosanitary desinfection against D. brevipes in pineapple. However, I have some concerns to be addressed, which are given below.
line 45. This information is not essential for the context of the introduction, but the data for that work is old. It would be necessary to update the information. FAO statistics can be consulted. From 2020 to 2022, the largest importers of pineapple have been the USA, China, China Mainland and Japan.
line 46-47. (Dysmicoccus brevipes) --> (Dysmicoccus brevipes (Cockerell, 1893)(Hemiptera: Pseudococcidae)
line 115. in [39] --> in Ren and Lee (2011) [39]
line 120. in [40] --> in Kwon et al. (2021) [40]
line 123. on [39] --> on Ren and Lee (2011) [39]
line 142. the formula [41] --> the Abbot formula [41]
line 284. reference 39 in not about ethyl formate
line 294. For future work, it may be interesting to repeat these experiments with the 20% pineapple loading ratio, because although it has been tested at doses of 15 and 30%, that intermediate value is estimated.
linea 321. 2 wk --> 2 weeks
line 370 and 381. References 6 and 11 are the same. The reference numbers in the text and bibliography must be modified.
REFERENCES. Some references appear with doi and others do not. The doi should be added to all articles that are available.
line 452. Finny --> Finney
Author Response

(The authors gave the same response as above.)
